# An investigation of how perceived smart tourism technologies affect tourists' well-being in marine tourism

**Yuxiang Zheng, Yue Wu**  *

School of Economics and Management, Shanghai Maritime University, Shanghai, China

* 202130710228@stu.shmtu.edu.cn

**Data Availability Statement:** All relevant data are within the paper and its Supporting Information files.

**Funding:** This work was supported by NATIONAL SOCIAL SCIENCE FUND OF CHINA PROJECT

## Abstract

Tourism industry is the first of the five happiness industries, playing a crucial role in enhancing people's well-being and happiness. Its high-quality development cannot be achieved without the use of emerging technologies, and today people have greatly improved the quality and happiness of tourism through smartphones, artificial intelligence, virtual reality and other technologies. Building smart marine tourism also requires widespread use of smart tourism technology. The study aims to examine the implications of perceived smart tourism technologies for tourist well-being in marine tourism, as well as the mediating role memorable tourism experiences play. We collected 445 valid questionnaires through a combination of offline and online methods and developed a theoretical model based on the results. The SPSS 26 statistical software package and Amos were applied in data analysis. There is a significant positive impact of perceived smart tourism technology on both hedonic and eudaimonic well-being of tourists in marine tourism, which is partially mediated by memorable tourism experiences. This paper provides certain suggestions and insights into the construction of smart marine tourism, so that managers can pay more attention to the experience and well-being of tourists and build a humanized, diversified, intelligent and innovative marine tourism. For smart tourism technology suppliers, it can provide them with new ideas for technological improvement, so that they can provide better services and attract more tourists in a market-oriented environment.

## Introduction

As technology continues to evolve, smart tourism technologies are increasingly used in different tourism areas [1, 2]. For example, AR and 3D maps are used effectively in mobile libraries in the context of e-learning [3]. Smart hotels are a new concept in the service industry [4], virtual avatars are also beginning to play an important role in digital hospitality [5]. Under the influence of COVID-19, tourists are increasingly inclined to use their smartphones to book transport and pay for shopping [6]. In addition, smart tourism technology is being used in smart museums and mega-events [7, 8]. Smart tourism is using information and communication technologies to facilitate tourism destinations for tourists and stakeholders such as suppliers [9, 10]. Smart tourism technologies encompass a wide range of tools and platforms,

[grant number 19BJY208]. Specifically, funders provide assistance in data collection.

**Competing interests:** The authors have declared that no competing interests exist.

including smart devices, social media platforms, cloud storage, data mining, the Internet of Things, AI algorithms, immersive VR and AR environments, NFC and RFID chips, all of which are utilized in various tourism activities. Among these technologies, VR and AR are rapidly emerging as key players in the smart tourism industry [11, 12]. Over the past few years, the tourism industry has witnessed a surge in the uptake and utilization of these technological advancements. Smart tourism destinations leverage big data to deliver personalized services that cater to user preferences, elevate the overall tourism experience, and boost the destination's competitiveness by leveraging smart tourism technologies [13]. As people increasingly prioritize their quality of life and happiness, the tourism industry has become highly competitive. Consequently, creating memorable tourism experiences has emerged as a crucial concept in recent academic research on tourism experiences. Pine et al. [14] proposed that the ability to craft unforgettable experiences for consumers is the cornerstone of success in the experience economy. In a study conducted by Jeong and Shin [15] and Hanna Lee [10] across various cities, it was demonstrated that the implementation of smart tourism technologies can lead to memorable tourism experiences and enhance tourist well-being. However, most studies on memorable tourism experiences have been directed at their effects on tourist satisfaction and behavioral intentions. There are fewer studies on the impact of memorable tourism experiences on tourists' well-being. Although hedonic well-being with happiness and pleasure as the main purpose is more common in the tourism literature, only a few tourism studies have used eudaimonic well-being, which is more concerned with the realization of full personal potential, virtues, character strengths, self-growth, self-fulfillment [16]. Considering the conceptual connotations of well-being purely in terms of the hedonic dimension does not provide insight into the various psychological states that tourists acquire as a result of participating in tourism activities, and eudaimonic dimensions related to well-being need to be considered [17]. Different types of tourism activities may bring different types of well-being to tourists [18], and there is no study that specifically examines the classification of tourists' well-being into hedonic and eudaimonic dimensions in marine tourism activities.

As one of the important components of tourism, marine tourism has become a core industry of the marine economy, which not only plays a significant role in boosting the local economy and facilitating the holistic growth of other sectors, but also fosters the overall advancement of coastal cities [19, 20]. China's marine tourism revenues are on the rise, and tourism serves as a crucial pillar of economic support for numerous coastal cities, so vigorously developing marine tourism can help targeted poverty alleviation [21, 22]. For South Asian countries, marine tourism is also crucial for sustainable economic development [23]. The advancement of living standards and the rise of the experience economy have presented both opportunities and challenges for the growth of marine tourism [24, 25]. However, there are still several issues that need to be addressed, such as the disconnection between services and tourists' needs and the lack of innovation in products and experiences [26].

In order to explore the gaps above, this paper proposes a theoretical framework, using literature combing and questionnaires, to examine how perceived smart tourism technology affects tourists' well-being in marine tourism, and to understand the underlying mechanisms of this impact. We explored the key factors affecting tourists' well-being, so as to improve the marine smart tourism technology experience and enhance visitor well-being.

## Literature review

### Perceived smart tourism technology

Scholars have already made contributions to identifying the attributes of perceived smart tourism technology. No and Kim [27] classified travel resources available on the internet into four

categories and identified five website attributes, with security being the main attribute of public websites. Huang et al. [12] identified the characteristics of smart tourism technology experiences as including informativeness, accessibility, interactivity, and personalization. In the context of websites, accessibility pertains to the ease with which users can obtain and utilize the information and services offered. This attribute is used to address the ease with which customers can locate and access the information and services offered by online information resources [28]. Security refers to the level of trustworthiness demonstrated by a website in safeguarding users' personal information [27]. Informativeness pertains to the accuracy, reliability and quality of the information obtained through smart tourism technologies [12]. Interactivity is a characteristic that enables prompt action, such as offering real-time feedback and engaging in active communication, including the sharing of information or ideas between users and administrators [27]. Personalization represents the capability of tourists to obtain precise and relevant information that caters to their travel planning requirements through the use of smart travel technology [27].

Research on perceived smart tourism technology in the literature primarily examines the effects of smart tourism technology on tourists, residents of tourist destinations, and tourism-related businesses. Technological advancements in smart tourism are accelerating, the way tourists travel has changed dramatically with the introduction of new channels such as social media and apps for booking, transportation, accommodation, and dining. Garcia-Milon et al. [6] examines how the COVID-19 pandemic moderates tourists' inclination to use smartphones while shopping, suggesting that barriers to using smartphones as a payment tool may have been reduced because electronic payment systems were preferred, even mandatory, during the epidemic. The COVID-19 pandemic has necessitated the utilization of robotics and other automated technologies by companies, and in the context of smart hotels, the implementation of innovative and advanced technologies should prioritize the needs and preferences of customers, rather than solely focusing on the technology itself [4].

In summary, this paper defines perceived smart tourism technology as the downloading or use of mobile applications or web pages (including transport, booking, reservations, maps, buying tourism products, etc.), the use of WeChat applets or QR codes, touch screen displays, NFC, VR experiences, AR experiences, etc. The perceived smart tourism technology referred to in this paper includes the pre-, mid- and post-tourism periods, and the five attributes of measuring perceived smart tourism technology are: security, informativeness, accessibility, interactivity and personalization.

## Memorable tourism experiences

Pizam [29] believed that the tourism and hospitality industry relies heavily on the creation of memorable experiences for tourists. The study of tourism experience satisfaction or quality of experience alone is not sufficient to adequately reflect the tourism experience sought by tourists, and the provision of high quality, memorable tourism experiences for tourists has become an important part of current tourism experience research [30, 31]. Larsen [32] first introduced the independent concept of tourism memory, arguing that the tourism experience is divided into three main parts: expectations, perceptions and memories, and that on-site tourism experiences are relatively ephemeral, while those that are recalled are the most important ones. Tung and Ritchie [30] conducted in-depth interviews with university students who had experienced travel and identified four basic dimensions of memorable travel experiences based on a grounded theory approach: affect, expectations, consequentiality and recollection. Jong-Hyeong Kim et al. [33] developed a scale to measure MTEs consisting of 24 questions on seven dimensions. Hedonism is a pleasurable feeling; refreshment refers to the feeling of self-renewal

experienced by the tourist as a result of moving from everyday life and stressful situations to the world of tourism; the term "local culture" encompasses the favorable perceptions that tourists have of the local population, as well as the opportunity to experience the local culture up close; meaningfulness refers to the sense of significance that the tourist has as a result of doing something important and worthwhile; knowledge pertains to the cognitive aspects such as gaining new insights, learning, and being educated; involvement refers to the level of interest and engagement exhibited by visitors towards a particular activity; and novelty stems from the newness that comes with a new experience. Research has shown that as travelers are always looking to explore new destinations, despite being highly satisfied with a destination, visitors may not choose to revisit it., and only those destinations that provide visitors with a memorable travel experience will attract more repeat visitors [34].

In summary, memorable tourism experiences are those that are remembered in a positive way and trigger memories even after the tour is over, and are formed by individuals based on their assessment of the tourism experience, resulting in experience memories that are selectively retained [35, 36]. Tourism experiences and memorable tourism experiences are not always synonymous, as not all tourism experiences have the potential to become truly unforgettable.

## Tourists' well-being

Research on tourists' well-being began in Lounsbury and Hoopes ' s [37] research on tourist life satisfaction. Happiness-related research in tourism is only 30 years old. The definition of the concept of tourist well-being has been influenced by philosophy and psychology, and the terms subjective well-being, psychological well-being, hedonic well-being and eudaimonic well-being have been used extensively in academic circles. The concept of tourists' well-being encompasses their emotional state of contentment and joy throughout the various stages of tourism, including the pre-trip planning, on-site activities, and post-trip reflection. The way in which tourist well-being is conceptualized in tourism has a significant impact on how destinations strive to improve the overall experience for visitors, as well as how the psychological advantages of tourism are perceived and evaluated. Filep's conclusion highlights the assessment of tourists' well-being throughout the three main stages of the tourism experience: anticipation, on-site, and reflection. Tourist well-being is characterized by positive emotions such as joy, interest, satisfaction, and love, as well as active participation in holiday activities and the ability to derive meaning from them [38]. Research has shown that tourism contributes to the well-being of social tourists, and the impact of tourism on social well-being is especially pronounced in certain areas, such as psychological resources, leisure pursuits, and family life, with tourists experiencing positive emotions through the tourism experience, leading to a sense of well-being [39, 40]. Using tourists in Xiamen as the study population, Su and Huang et al. [41] investigated how service quality and fairness affect tourist behavior and subjective well-being. The results prove that service fairness affects tourists' subjective well-being more than service quality. Using South African tourists as the subject of the study, Saayman and Li et al. [42] examined the relationship between the tourism experience and tourist satisfaction, as well as the subsequent impact on their overall well-being, showing that satisfaction with destination services significantly affects the subjective well-being of travelers. Sebastian Filep [43] argued for the need for a comprehensive qualitative assessment of tourism experiences that hold significant meaning and, as well as those special and captivating moments that leave a lasting impression on tourists, highlighting the problems with conceptualizing visitor well-being as subjective well-being: subjective well-being struggles to explain meaningful holiday experiences and participation in live experiences. The argument is that a complete understanding of well-being should incorporate theoretical constructs such as positive affect, engagement, and a

sense of significance and direction in life, which provide a more comprehensive explanation of the connection between tourist well-being and the tourism experience than subjective well-being alone. Park and Ahn [44] conducted a survey of tourists who had travel experiences abroad or on Jeju Island within a year and showed that pleasure and transcendental experiences had a beneficial impact on the hedonic well-being of travelers, while personal meaning and self-reflective experiences had a positive influence on eudaimonic travel well-being.

The conceptualization of happiness is heavily influenced by hedonic theory, which defines well-being in terms of the pursuit of maximal pleasure and the minimization of pain, emphasizing positive emotions such as enjoyment, pleasure and joy. However, considering the conceptual connotations of happiness purely in terms of the hedonic dimension does not provide insight into the various psychological states that tourists acquire as a result of participating in tourism activities, and non-hedonic dimensions related to well-being need to be considered. The eudaimonic well-being focuses on the full development of one's potential, virtues, character strengths, self-growth and self-fulfillment. In this paper, hedonic and eudaimonic well-being are used to measure tourists' well-being.

## Marine tourism

Marine tourism has brought varying degrees of economic growth to coastal areas, which has broad prospects for development [19–21]. The innovation of marine tourism products plays a crucial role in the advancement of traditional tourism [25]. Without product innovation and optimization, it is impossible to create marine tourism products that cater to the demands of tourists, thus hindering the development of marine tourism resources. Even if more tourism resources are developed, they will not bring consumers a sense of participation and freshness [45]. In recent years scholars have been conducting more and more in-depth research on marine tourism. Xu [26] gives strategies to cope with the lack of innovation in marine tourism by analysing the RMP model. Qiu [46] gives the promotion strategy of tourism cultural and creative products in the era of Internet+. Research has shown that understanding the cognitive dimensions of the marine tourism experience will result in a more memorable experience for the tourists and may lead to post-purchase behaviors such as repeat purchases and word-of-mouth referrals, which in turn will increase the commercial interests [47]. Li and Yu [48] take Zhuhai marine tourism as the research object to quantify and optimize the influencing factors of marine tourism economic development. By exploring the integrated development of Japan's marine fisheries and marine tourism industries, Yao and Zhang et al. [49] give related advices for promoting the transformation and upgrading of the marine industry and industrial convergence. In addition, some scholars have provided new development ideas for marine sports tourism based on the perspective of experience economy [24, 50, 51]. Current marine tourism is becoming more and more intelligent and informative [52]. Technological innovation is indispensable for the construction of smart marine tourism. Internet and IoT based marketing models have been widely used in the marine tourism industry [52, 53]. In recent years, there has been a growing acceptance and popularity of virtual ocean products among individuals. Consequently, the design of virtual ocean products that rely on visual communication has emerged as a prominent and challenging area of research [54]. Zhu and Hou et al. [55] introduced a marine tourism geographic information visualization system that utilizes big data fusion and B/S architecture. By analyzing the characteristics of marine culture and blue ocean technology, Chen and Zhou [56] explored the significance of intelligent marine tourism product packaging for establishing brand image in the context of Internet.

To sum up, traditional marine tourism has been developing in the direction of smart marine tourism, and the support of new technology and accurate catering to consumer needs are indispensable.

## Hypotheses development

### Proposed hypothesis regarding the correlation between perceived smart tourism technology and tourists' well-being

The integration of smart tourism technologies and the creation of unforgettable tourism experiences are crucial factors in enhancing tourist satisfaction and fostering destination loyalty [57]. Research has shown that the use of smart tourism technology can enhance memorable travel experiences, which in turn increases tourist satisfaction and revisit intentions [58]. The perceived attributes of smart tourism technology—specifically, informativeness, interactivity, and personalization—are crucial factors that influence tourist experience and the likelihood of revisiting a destination [15]. Pai and Liu et al. [59] concluded that smart tourism technologies can improve tourism experience satisfaction and thus enhance tourist happiness. Pai, Kang et al. [60] conducted a survey with tourists who had used local smart tourism technology in Macau and the study's findings indicated that the perceived use of smart tourism technology had a notable and positive effect on the enjoyment of travel. The results of a questionnaire survey of 191 foreign tourists from pairs of Seoul, South Korea, showed that perceived smart tourism technology and perceived value of destination significantly influenced tourists' overall well-being [10]. Smart tourism technologies assist tourists in creating a feeling of excitement when exploring a destination, while also alleviating concerns about uncertainty in the pre-trip planning process [61]. Jia and Zhang [62] found that virtual tourism products provide objective tourism situations characterized by simulation, richness and creativity, which play a role in different stages of the tourism experience and intervene in the process of generating and precipitating tourist well-being. Tuo and Qin [63] propose that the use of AI technology can enhance tourist well-being in the following ways: (1) to leverage the healing function of AI technology to enhance the positive emotions of tourists (2) to promote social connectivity (3) to create an immersive tourism experience (4) to enhance the sense of control of tourists (5) to drive the co-creation of value among tourists to achieve an achievement experience and personal growth. Digital technology promotes the well-being of travelers and residents of tourist destinations by facilitating the development of high-quality tourism and facilitating the interaction between hosts and guests [64]. Tourism, with its highly experiential, spiritual and cultural characteristics, is becoming an important application scenario for cutting-edge technologies. The development of tourism improves people's sense of well-being, and the application of smart tourism technologies creates a more convenient, enjoyable and memorable travelling experience for tourists by expanding the destination co-creation space. However, the negative impacts of the technology cannot be ignored, such as the privacy crisis and the challenge of human-computer trust [63]. Therefore, it is necessary to study the mechanism of the impact of perceived smart tourism technology on tourists' well-being. In light of this, the following hypotheses were developed:

H1: There is a significant positive relationship between perceived smart tourism technology and tourist well-being

### Proposed hypothesis regarding the correlation between perceived smart tourism technologies and memorable tourism experiences

The idea of providing tourists with unique and unforgettable experiences is a prevalent concept within the tourism industry. As destination choice increases and competition intensifies, it is essential for destinations to discover creative methods to distinguish themselves and provide visitors with experiences that offer unparalleled value. The advent of information and communication technologies has given way to the emergence of smart tourism, where the

tourism sector has embraced the use of smart technologies. Through the introduction and conceptualization of a novel experience creation paradigm known as "technology-enhanced destination experiences", a space for creating memorable experiences for tourists is created. The proposal involves the expansion of destination experience co-creation to encompass the pre-, during-, and post-tourism phases [65]. Jeong and Shin [15] carried out a research targeting visitors to the five most prominent smart cities in the United States, the results show that the use of smart tourism technology by travelers to access information about their tourism activities, and interact with tourism staff has been found to contribute to the creation of memorable tourism experiences. Furthermore, Hanna Lee [10] argues that visitors' access to information through smart tourism technology helps them streamlines the decision-making process and improves their tour experience, which would emerge as a significant factor in enhancing a memorable tour experience. The use of smart tourism technology has been found to have a positive influence on the creation of memorable tourism experiences for visitors to museums, an impact that surpasses conventional offerings like museum displays and personnel assistance, and smart tourism technology can be seen as a stand-alone new efficient service for museums rather than a complement to existing services [7]. The tourism industry is benefiting from the use of smart tourism technologies, which are helping to create successful marketing plans to draw in more tourists and offer them exceptional travel experiences. Accordingly, we propose the following hypotheses:

H2: There is a significant positive relationship between perceived smart tourism technology and memorable tourism experiences

## Hypothesis of the mediating role of memorable tourism experiences

Based on empirical research, it has been found that tourism is a pursuit of pleasurable experiences, and the level of satisfaction that tourists derive from their travels is influenced by factors such as their personality, the type of destination they choose, and the kind of tourism activities they engage in [43, 66, 67]. Travel has been found to have a positive impact on well-being by mitigating the effects of hedonic adaptation, particularly in terms of managing expectations and encountering unexpected discoveries [68]. Positive experiences during travel have the potential to enhance individuals' well-being, and social interaction can be deemed as the foremost factor in enhancing well-being [69]. One study investigated the value-seeking process of tourists and the results showed that the happiness of tourists can be enhanced by their contentment with the overall travel experience as well as their satisfaction with the quality of service provided [10]. Research has shown that memorable tourism experiences are significantly associated with tourist well-being [70]. Experiencing pleasurable and memorable travel moments can contribute to both hedonic and eudaimonic well-being [71]. According to Matteucci and Filep's [72] research, participating in flamenco music and dance workshops was found to have a significant positive impact on well-being, particularly in terms of fostering self-actualization and personal growth. Holidays have the potential to increase the level of well-being of those who enjoy them, leading to hedonic well-being [73]. The novelty quest of a memorable tourism experience is also recognized to have a substantial influence on subjective well-being, either in terms of their positive emotions and happiness or their overall life satisfaction [74]. In addition, Li and Chan [75] found that overseas Chinese travelling back to their home countries helps create meaning and life purpose, which leads to happiness. Accordingly, we propose the following hypotheses:

H3: There is a significant positive relationship between memorable tourism experiences and tourists' well-being

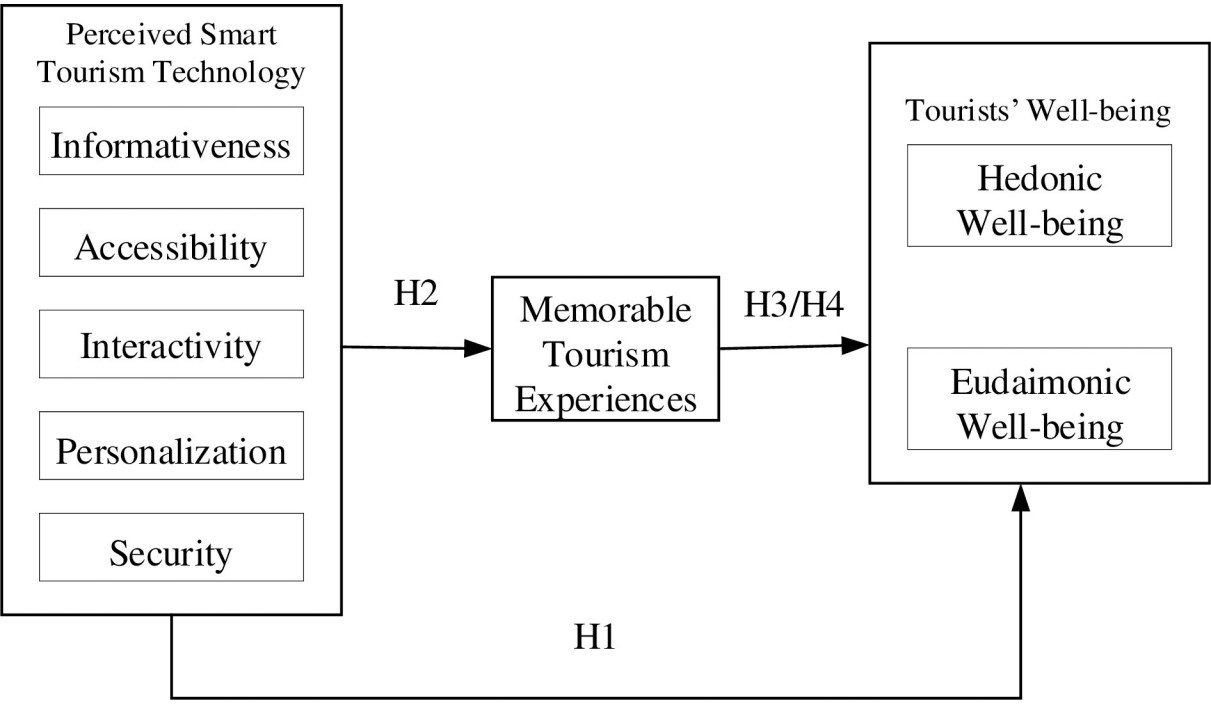

**Fig 1. Proposed model.**

H4: The connection between perceived smart tourism technology and tourists' well-being is mediated by memorable experiences during their travels

(**Fig 1**) illustrates the proposed relationships

## Research method

### Measurement development

This paper draws on previously validated scales in the context of the actual study, with appropriate modifications to each question item. We measured all question items using a Likert scale of 1 to 5 (strongly disagree to strongly agree). Smart tourism technology was divided into five dimensions to measure question items drawing from Hanna Lee et al. [10, 12, 27, 76]. The scale developed by Jeong and Shin et al. [15, 77] was used to measure memorable travel experiences in context. Tourists' well-being is divided into two dimensions, and the measurement questions are based on research conducted by Diener et al. [78, 79]. Table 1 shows the specific question items.

### Data collection

This study is aimed at people who have experienced marine tourism and have experienced smart tourism technology. We used a combination of online and offline methods to collect empirical data. An offline questionnaire survey was conducted with visitors to Shanghai Changfeng Ocean World, Shanghai Ocean Aquarium and Shanghai Haichang Ocean Park. Family members, classmates and friends were asked to help with snowballing research and questionnaires were distributed through a social networking platform—WeChat. We selected respondents using the following two screening questions: firstly, have you ever experienced marine tourism, and secondly, have you experienced smart tourism technology on this trip?

**Table 1. Questionnaire items.**

| Construct | Item | Item description |
|---|---|---|
| Perceived Smart Tourism Technology | Informativeness (INF) | |
| | INF1 | Tourism websites and apps provide me with useful information about the travel destination(s) and the trip |
| | INF2 | Tourism websites and apps are helpful for evaluating the destination(s) and the trip |
| | INF3 | Tourism websites and apps enable me to complete my trip with the detailed information provided |
| | INF4 | Tourism websites and apps enable me to minimize my worries about my trip |
| | Accessibility (ACC) | |
| | ACC1 | I can use smart tourism technologies such as tourism websites and apps anytime and anywhere |
| | ACC2 | I can easily find smart tourism technologies such as tourism websites and apps |
| | ACC3 | I can easily use smart tourism technologies such as tourism websites and apps |
| | ACC4 | I can search without a complicated sign-up process at tourism websites and apps |
| | Interactivity (INT) | |
| | INT1 | I can find many other travelers' questions and answers on tourism websites and apps |
| | INT2 | Smart tourism technologies such as tourism websites and apps that I use are highly responsive to me |
| | INT3 | It is easy to share tourism information content on tourism websites and apps |
| | Personalization (PER) | |
| | PER1 | Smart tourism technologies such as tourism websites and apps allow me to receive tailored information |
| | PER2 | I can interact with smart tourism technologies such as tourism websites and apps to get personalized information |
| | PER3 | The tourism information provided by smart tourism technologies such as tourism websites and apps meets my needs |
| | Security (SEC) | |
| | SEC1 | I am not concerned that too much personal information is collected when I use smart tourism technologies |
| | SEC2 | I have no doubt that my privacy is protected well when I use smart tourism technologies such as tourism websites and apps |
| | SEC3 | I am not concerned with the security of sensitive information when I use smart tourism technologies such as tourism websites and apps. |
| Memorable Tourism Experiences (MTE) | MTE1 | I really enjoyed this tourism experience |
| | MTE2 | I revitalized through this tourism experience |
| | MTE3 | I learned something about myself from this tourism experience |
| | MTE4 | I had a chance to closely experience the local culture of a destination area |
| | MTE5 | I experienced something new (e.g., food, activity, etc.) during this tourism experience |
| | MTE6 | My experience with using smart tourism technologies was unforgettable. |
| Tourist Well-being | Hedonic Well-being (HWB) | |
| | HWB1 | In general, I consider myself very happy |
| | HWB2 | Compared to most of my peers, I consider myself happier |
| | HWB3 | I am generally very happy and enjoy life |
| | HWB4 | In most ways my life is close to my ideal |
| | HWB5 | I'm satisfied with my life |
| | Eudaimonic Well-being (EWB) | |
| | EWB1 | I can resist social pressures to think and keep my opinions |
| | EWB2 | I feel I am in charge of the situation in which I live |
| | EWB3 | I have a feeling of continued development; I think I'm growing |
| | EWB4 | I like most aspects of my personality |
| | EWB5 | I have warm, satisfying, and trusting relationships with others |

(*Continued*)

**Table 1.** (Continued)

| Construct | Item | Item description |
|---|---|---|
| | EWB6 | I have a sense of purpose in my life |

INF: Informativeness; ACC: Accessibility; INT: Interactivity; PER: Personalization; SEC: Security; MTE: Memorable Tourism Experiences; HWB: Hedonic Well-being; EWB: Eudaimonic Well-being

The questionnaires are anonymous. We don't have access to information that could identify individual participants during or after data collection. Within three weeks (from 1 March 2023 to 22 March 2023), we received 418 questionnaires and screened 401 valid ones. Over 45% of the respondents were between the ages of 18 and 30. Table 2 lists more details of the sample for this study.

## Data analysis and results

### Measurement model

To evaluate the reliability and validity of the study, Confirmatory Factor Analysis (CFA) was conducted using maximum likelihood estimation. The findings revealed that the model fit indices were satisfactory and met the acceptable standards: (CMIN/DF = 1.157, GFI = 0.990, AGFI = 0.915, CFI = 0.990, IFI = 0.990, TLI = 0.989, RMSEA = 0.019, SRMR = 0.026). Each variable had an average variance extracted (AVE) score of above 0.40, which is considered acceptable by Fornell and Larcker's standards for AVE between 0.36–0.5 [80]. The composite reliability (CR) score for each variable was above 0.70, which is deemed acceptable for all the

**Table 2.** Sample profile of the respondents.

| Dimension | Items | Frequency | (%) |
|---|---|---|---|
| Gender | Male | 190 | 47.4 |
| | Female | 211 | 52.6 |
| Age (years) | 18–30 | 181 | 45.1 |
| | 31–40 | 107 | 26.7 |
| | 41–50 | 70 | 17.5 |
| | 51–60 | 34 | 8.5 |
| | >60 | 9 | 2.2 |
| Education level | Junior high school or lower | 21 | 5.2 |
| | Senior or vocational high school | 41 | 10.2 |
| | Junior college | 86 | 21.4 |
| | Bachelor | 186 | 46.4 |
| | Master or higher | 67 | 16.7 |
| Monthly income(yuan) | ≤4000 | 25 | 6.2 |
| | 4001–6000 | 234 | 58.4 |
| | 6001–8000 | 110 | 27.4 |
| | ≥8001 | 32 | 8.0 |
| Occupation | Military servants, civil workers, and teachers | 45 | 11.2 |
| | Industrial and business industries | 243 | 60.6 |
| | Freelancers | 82 | 20.4 |
| | Retired | 9 | 2.2 |
| | Student | 11 | 2.7 |
| | Others | 11 | 2.7 |

**Table 3. Measurement model.**

| Constructs | Items | Factor loadings | Cronbach's a | CR | AVE |
|---|---|---|---|---|---|
| Informativeness (INF) | INF1 | 0.988 | 0.833 | 0.8497 | 0.5933 |
| | INF2 | 0.723 | | | |
| | INF3 | 0.679 | | | |
| | INF4 | 0.643 | | | |
| Accessibility (ACC) | ACC1 | 0.69 | 0.750 | 0.75 | 0.4289 |
| | ACC2 | 0.632 | | | |
| | ACC3 | 0.633 | | | |
| | ACC4 | 0.663 | | | |
| Interactivity (INT) | INT1 | 0.683 | 0.741 | 0.7418 | 0.4893 |
| | INT2 | 0.72 | | | |
| | INT3 | 0.695 | | | |
| Personalization (PER) | PER1 | 0.725 | 0.771 | 0.7705 | 0.5281 |
| | PER2 | 0.717 | | | |
| | PER3 | 0.738 | | | |
| Security (SEC) | SEC1 | 0.703 | 0.727 | 0.7259 | 0.469 |
| | SEC2 | 0.687 | | | |
| | SEC3 | 0.664 | | | |
| Memorable Tourism Experiences (MTE) | MTE1 | 0.655 | 0.841 | 0.8423 | 0.4713 |
| | MTE2 | 0.717 | | | |
| | MTE3 | 0.672 | | | |
| | MTE4 | 0.671 | | | |
| | MTE5 | 0.728 | | | |
| | MTE6 | 0.673 | | | |
| Hedonic Well-being (HWB) | HWB1 | 0.67 | 0.805 | 0.8053 | 0.4527 |
| | HWB2 | 0.667 | | | |
| | HWB3 | 0.684 | | | |
| | HWB4 | 0.674 | | | |
| | HWB5 | 0.669 | | | |
| Eudaimonic Well-being | EWB1 | 0.697 | 0.820 | 0.818 | 0.4289 |
| | EWB2 | 0.593 | | | |
| (EWB) | EWB3 | 0.663 | | | |
| | EWB4 | 0.652 | | | |
| | EWB5 | 0.665 | | | |
| | EWB6 | 0.665 | | | |

INF: Informativeness; ACC: Accessibility; INT: Interactivity; PER: Personalization; SEC: Security; MTE: Memorable Tourism Experiences; HWB: Hedonic Well-being; EWB: Eudaimonic Well-being

factors. The factor loadings were both statistically significant and positive (Table 3). Convergent validity is demonstrated by significant loadings on all items on their corresponding constructs. The discriminant validity of the study was established as the square root of average variance extracted for each construct was higher than the correlation between the constructs. Furthermore, Table 4 presents the means and standard deviations (SD) of the study variables.

## Hypothesis testing

The results of the structural analysis (Table 5) suggests that the relationship between INF and HWB is positive and significant with the values of β = 0.165, t-value = 3.352, and p < 0.001;

Table 4. Discriminant validity of this study.

| Variables | Mean | SD | INF | ACC | INT | PER | SEC | MTE | HWB | EWB |
|---|---|---|---|---|---|---|---|---|---|---|
| INF | 3.8697 | 0.92155 | *0.77026* | | | | | | | |
| ACC | 3.8017 | 0.88476 | 0.61 | *0.654905* | | | | | | |
| INT | 3.8644 | 0.88126 | 0.6 | 0.574 | *0.6995* | | | | | |
| PER | 3.9071 | 0.84858 | 0.559 | 0.524 | 0.53 | *0.726705* | | | | |
| SEC | 3.7678 | 0.92351 | 0.559 | 0.544 | 0.58 | 0.533 | *0.684836* | | | |
| MTE | 3.8397 | 0.85367 | 0.582 | 0.549 | 0.564 | 0.522 | 0.554 | *0.686513* | | |
| HWB | 3.8166 | 0.85244 | 0.566 | 0.548 | 0.558 | 0.504 | 0.522 | 0.539 | *0.67283* | |
| EWB | 3.8337 | 0.80899 | 0.572 | 0.547 | 0.559 | 0.532 | 0.553 | 0.551 | 0.54 | *0.654905* |

Please note that in the table, the italicized data indicate the square root of the average variance extracted for each construct. The off-diagonal values represent the correlations between the latent variables. INF: Informativeness; ACC: Accessibility; INT: Interactivity; PER: Personalization; SEC: Security; MTE: Memorable Tourism Experiences; HWB: Hedonic Well-being; EWB: Eudaimonic Well-being

the relationship between ACC and HWB is positive and significant with the values of $\beta = 0.235$, t-value = 5.137, and $p < 0.001$; the relationship between INT and HWB is positive and significant with the values of $\beta = 0.271$, t-value = 5.928, and $p < 0.001$; the relationship between PER and HWB is positive and significant with the values of $\beta = 0.150$, t-value = 3.564, and $p < 0.001$; the relationship between SEC and HWB is positive and significant with the values of $\beta = 0.125$, t-value = 3.564, and $p < 0.001$.

The relationship between INF and EWB is positive and significant with the values of $\beta = 0.210$, t-value = 4.479, and $p < 0.001$; the relationship between ACC and EWB is positive and significant with the values of $\beta = 0.164$, t-value = 3.762, and $p < 0.001$; the relationship between

Table 5. Regression weights between the proposed relationships.

| Relationship | Standardized path coefficient (β) | t-value | R$^2$ | Hypothesis |
|---|---|---|---|---|
| INF→HWB | 0.165 | 3.352*** | 0.714 | Supported |
| ACC→HWB | 0.235 | 5.137*** | | Supported |
| INT→HWB | 0.271 | 5.928*** | | Supported |
| PER→HWB | 0.150 | 3.564*** | | Supported |
| SEC→HWB | 0.125 | 3.069*** | | Supported |
| INF→EWB | 0.210 | 4.479*** | 0.741 | Supported |
| ACC→EWB | 0.164 | 3.762*** | | Supported |
| INT→EWB | 0.211 | 4.838*** | | Supported |
| PER→EWB | 0.211 | 5.287*** | | Supported |
| SEC→EWB | 0.180 | 4.660*** | | Supported |
| INF→MTE | 0.266 | 6.059 *** | 0.774 | Supported |
| ACC→MTE | 0.136 | 3.343 *** | | Supported |
| INT→MTE | 0.230 | 5.651 *** | | Supported |
| PER→MTE | 0.166 | 4.444 *** | | Supported |
| SEC→MTE | 0.193 | 5.326 *** | | Supported |
| MTE→HWB | 0.798 | 27.702*** | 0.653 | Supported |
| MTE→EWB | 0.818 | 29.390*** | 0.676 | Supported |

Notes: ***$p < 0.001$, ***$p < 0.010$, *$p < 0.050$

INF: Informativeness; ACC: Accessibility; INT: Interactivity; PER: Personalization; SEC: Security; MTE: Memorable Tourism Experiences; HWB: Hedonic Well-being; EWB: Eudaimonic Well-being

**Table 6. Results for mediation testing.**

| Relationship | Mediator-free direct effect | Mediator-direct effect | Indirect effect |
|---|---|---|---|
| INF→MTE→HWB | 0.165 ** | 0.087 | Full mediation |
| ACC→MTE→HWB | 0.235 *** | 0.195 *** | Partial mediation |
| INT→MTE→HWB | 0.271 *** | 0.204 *** | Partial mediation |
| PER→MTE→HWB | 0.150 *** | 0.101 * | Partial mediation |
| SEC→MTE→HWB | 0.125 ** | 0.068 | Full mediation |
| INF→MTE→EWB | 0.210*** | 0.133** | Partial mediation |
| ACC→MTE→EWB | 0.164*** | 0.125** | Partial mediation |
| INT→MTE→EWB | 0.211*** | 0.144** | Partial mediation |
| PER→MTE→EWB | 0.211*** | 0.163*** | Partial mediation |
| SEC→MTE→EWB | 0.180*** | 0.125** | Partial mediation |

Notes

* * *p < 0.001, * * *p < 0.010

*p < 0.050

INF: Informativeness; ACC: Accessibility; INT: Interactivity; PER: Personalization; SEC: Security; MTE: Memorable Tourism Experiences; HWB: Hedonic Well-being; EWB: Eudaimonic Well-being

INT and EWB is positive and significant with the values of $\beta = 0.211$, t-value = 4.838, and $p < 0.001$; the relationship between PER and EWB is positive and significant with the values of $\beta = 0.211$, t-value = 5.287, and $p < 0.001$; the relationship between SEC and EWB is positive and significant with the values of $\beta = 0.180$, t-value = 4.660, and $p < 0.001$, hence H1 is supported.

The relationship between INF and MTE is positive and significant with the values of $\beta = 0.266$, t-value = 6.059, and $p < 0.001$; the relationship between ACC and MTE is positive and significant with the values of $\beta = 0.136$, t-value = 3.343, and $p < 0.001$; the relationship between INT and MTE is positive and significant with the values of $\beta = 0.230$, t-value = 5.651, and $p < 0.001$; the relationship between PER and MTE is positive and significant with the values of $\beta = 0.166$, t-value = 4.444, and $p < 0.001$; the relationship between SEC and MTE is positive and significant with the values of $\beta = 0.193$, t-value = 5.326, and $p < 0.001$, hence H2 is supported.

The relationship between MTE and EWB is positive and significant with the values of $\beta = 0.798$, t-value = 27.702, and $p < 0.001$, and the relationship between MTE and HWB is positive and significant with the values of $\beta = 0.818$, t-value = 29.390, and $p < 0.001$, hence H3 is supported. The mediation results are shown in Table 6, H4 is supported.

## Discussion and conclusion

The key outcomes of this research indicate that the perceived smart tourism technology has a notable influence on both hedonic and eudaimonic aspects of well-being, with interactivity having primary influence on tourists' hedonic well-being, probably because interactivity refers mainly to the ability of smart tourism technologies to provide a platform for communication and interaction between tourists and tourists, and between tourists and the destination, with tourists preferring more interactive experiences [81]. Because smart tourism technologies enable real-time dialogue through direct messaging, live feedback mechanisms or customer feedback, tourists who engage in substantial online interactions tend to have a more favorable experience and higher hedonic well-being than those with little or limited Internet interaction.

Personalization has the greatest impact on visitors' eudaimonic well-being, possibly because it lies in providing visitors with tailored information that is more conducive to their self-worth.

According to this study, memorable tourism experiences were significantly impacted by perceived smart tourism technology, with informativeness being the most significant factor influencing memorable tourism experiences. This finding is consistent with earlier studies in the field that information technology contributes to tourism and therefore high quality information is vital [27]. The term information quality pertains to the level of correspondence between the information provided and the information needed by the individual [82]. For example, Jeong and Shin [15] show that informativeness is a key factor affecting the memorable experience of visitors as smart tourism cities are equipped with high-speed network capacity to its full extent.

The findings of this study indicate that tourists can experience an improvement in both their hedonic and eudaimonic well-being through the creation of memorable travel experiences. Memorable tourism experiences can lead to short-term hedonic well-being among visitors, as well as long-term increases in self-worth and eudaimonic well-being. Existing empirical research supports this result. For example, studies have found that flamenco tourism experiences in Spain contribute to feelings of self-fulfillment and satisfaction [72]. Tourism can be viewed as a market that offers a diverse range of experiences to consumers and tourists do not just buy products and services, they want memorable experiences [83]. The degree of meaningfulness and memorability of the tourist experience plays a crucial role in determining whether a trip contributes to both hedonic and eudaimonic well-being. In addition, this study found that the impact of perceived smart tourism technology on tourists' well-being is partly mediated by the creation of memorable tourism experiences.

Based on the above results and discussion, focusing on the use of smart tourism technology, especially the informativeness, interactivity and personalization, and doing our best to create memorable marine tourism experience and well-being for tourists is undoubtedly one of the effective ways to accelerate the construction of smart marine tourism.

## Implications

### Theoretical contributions

This research investigates the influence of perceived smart tourism technology on tourists' well-being in the context of marine tourism, as well as the mediating effect of memorable tourism experiences on this association. Firstly, this paper examines the impacts of perceived smart tourism technology on tourist well-being and contributes to the enrichment of the literature in perceived smart tourism technology. Furthermore, this research makes a significant contribution to the field of positive psychology by building on existing research and examining two distinct dimensions of tourist well-being, namely hedonic well-being and eudaimonic well-being, which, although different in nature, are not mutually exclusive [84]. Finally, this study strengthens existing research by finding that memorable tourism experience is positively associated with tourist well-being and serves as a partial mediator in the relationship between perceived smart tourism technologies and tourist well-being.

### Managerial implications

Firstly, smart destinations can improve the quality of information, disseminate more useful and timely information [12], improve the usability and ease of use of smart tourism technology, make it convenient for visitors to enjoy anytime and anywhere [28], enhance interactive experiences and improve the fun of visiting, while protecting visitors' personal privacy and enhancing personalized services [27]. Secondly, managers should be aware that memorable

tourism experiences are not limited to a specific situation, but vary from person to person, and that tourism providers should pay due attention to the differences in tourist experiences [85]. Finally, tourism businesses can utilize well-being as a crucial resource to promote their products and services, using it as a marketing tool to influence consumers' decisions on their choice of travel destination [70]. Tourist well-being is not only reflected in short-term hedonic well-being, but also in eudaimonic well-being related to personal growth and self-worth realization [85]. By implementing a philosophy of well-being in tourism destinations, more people can be encouraged to understand and engage in tourism, which in turn can play a significant role in the recovery and revitalization of the tourism industry in the post-epidemic era.

## Limitation and future research

This study employed a questionnaire as the primary method of data collection, with a valid sample of 401, the largest proportion being young people. In the future, a more refined study of tourist experience and well-being could be conducted for a specific group, such as older people or children. It is important to note that the scope of this research is confined to the specific domain of marine tourism, which does not adequately represent all tourism scenarios, and the sample size and research context could be expanded in the future for a more adequate study. This paper takes the research of existing scholars to explore perceived smart tourism technology in five dimensions, and future research can delve deeper into other aspects of perceived smart tourism technology.

## Supporting information

**S1 Data.**
(RAR)

## Author Contributions

**Formal analysis:** Yue Wu.

**Writing – original draft:** Yuxiang Zheng.

**Writing – review & editing:** Yue Wu.

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
