## [Decision Letter · Decision Letter 0]

27 Jun 2023

PONE-D-23-12482An investigation of how perceived smart tourism technologies affect tourists' well-being in marine tourismPLOS ONE

Dear Dr. Wu,

Thank you for submitting your manuscript to PLOS ONE. After careful consideration, we feel that it has merit but does not fully meet PLOS ONE’s publication criteria as it currently stands. Therefore, we invite you to submit a revised version of the manuscript that addresses the points raised during the review process.

We look forward to receiving your revised manuscript.

Kind regards,

Sudarsan Jayasingh, Ph.D

Academic Editor

PLOS ONE

Journal Requirements:

2. You indicated that ethical approval was not necessary for your study. We understand that the framework for ethical oversight requirements for studies of this type may differ depending on the setting and we would appreciate some further clarification regarding your research. 

Could you please provide further details on why your study is exempt from the need for approval and confirmation from your institutional review board or research ethics committee (e.g., in the form of a letter or email correspondence) that ethics review was not necessary for this study? 

Please include a copy of the correspondence as an ""Other"" file.

**Additional Editor Comments:**

I find your studies involved children and hope the required ethical considerations are taken care. Theoretical and managerial contribution need to be improved by citing the pervious research done on this topic.

Reviewers' comments:

Reviewer's Responses to Questions

**Comments to the Author**

1. Is the manuscript technically sound, and do the data support the conclusions?

Reviewer #1: Partly

Reviewer #2: Partly

2. Has the statistical analysis been performed appropriately and rigorously? 

Reviewer #1: I Don't Know

Reviewer #2: Yes

3. Have the authors made all data underlying the findings in their manuscript fully available?

Reviewer #1: Yes

Reviewer #2: No

4. Is the manuscript presented in an intelligible fashion and written in standard English?

Reviewer #1: Yes

Reviewer #2: Yes

5. Review Comments to the Author

Reviewer #1: Overall, this study is interesting, and it is evident that the author has put effort into presenting the descriptive content of the investigation. I have a few questions I would like to discuss with the author:

The author mentions the application of smart tourism technology in different tourism domains. Could the author provide some specific examples through the literature? Why is there a special focus on marine tourism in the context of the tourism field? What is the main gap or rationale for this focus?

In the section discussing hypothesis testing, the author briefly mentions literature on the impact of smart tourism technology on tourist well-being. Could the author provide further elaboration on why they are exploring the relationship between these two factors? Did they consider other potential confounding variables during the process? For example, does the attractiveness of the destination or personal characteristics of tourists influence well-being?

Regarding the research on tourist well-being mentioned in the paper, is there any further explanation or differentiation of different types of well-being and their relationship with the tourist experience?

As mentioned in this study, previous tourism literature has already discussed the relationship between smart technology and well-being. This study specifically investigates marine tourism. However, in the hypothesis and results discussion, it is not clear why marine tourism was chosen and there is a lack of literature discussion and traceability regarding marine tourism. This might weaken the contribution of this article. It is recommended to strengthen the literature review and discussion related to marine tourism.

Reviewer #2: This research discusses how perceived smart tourism technologies affect tourists' well-being. However, some contents in the manuscript did not present logically, especially reviewer can hardly see the research gap and implications through the manuscript. Additionally, the method of data collection was unclear and some paragraphs are problematics. Therefore, the following comments are provided and hope it helps improve the quality of this manuscript.

Introduction

1. P3, “Although hedonic well-being with … personal potential, virtues, character strengths, self-growth, self-fulfillment”. The authors only describe which aspects it pays more attention to in the few tourism literatures that study eudaimonic well-being, but do not discuss the gaps between this study and existing research. Please see some of the examples below:

Li, T. E., & Chan, E. T. H. (2017). Diaspora tourism and well-being: A eudaimonic view. Annals of Tourism Research, 63, 205-206.

Hao, F., & Xiao, H. (2021). Residential tourism and eudaimonic well-being: A ‘value-adding’analysis. Annals of Tourism Research, 87, 103150.

Su, L., Tang, B., & Nawijn, J. (2020). Eudaimonic and hedonic well-being pattern changes: Intensity and activity. Annals of Tourism Research, 84, 103008.

Huang, X., Wang, P., & Wu, L. (2023). Well-being Through Transformation: An Integrative Framework of Transformative Tourism Experiences and Hedonic Versus Eudaimonic Well-being. Journal of Travel Research, 00472875231171670.

Rahmani, K., Gnoth, J., & Mather, D. (2018). Hedonic and eudaimonic well-being: A psycholinguistic view. Tourism Management, 69, 155-166.

Chang, L., Moyle, B. D., Dupre, K., Filep, S., & Vada, S. (2022). Progress in Research on Seniors' well-being in tourism: A systematic review. Tourism Management Perspectives, 44, 101040.

Literature review

2. P9-P10, “The integration of smart tourism technologies…. In light of this, the following hypotheses were developed”. The discussion on the relationship between perceived smart tourism technology and tourist well-being in this paragraph is relatively weak. The examples given by the authors do not well support the hypothesis.

Research method

3. P13-P14, “This paper draws on previously validated scales…are based on research conducted by Diener”. Reviewers cannot see the scale for this study. It is recommended to show the complete scale.

Data collection

4. P14, “This study is aimed at people…Table 1 lists more details of the sample for this study”. Why did the authors use a combination of online and offline methods to collect data? After collecting these two parts of the data, was there a t-test performed? How are offline questionnaires collected?

5. P14, Table1: It is inappropriate to collect questionnaires from participants younger than 18 years of age.

Data analysis and results

6. Table 2, Table 4 and Table 5: The authors did not explain what each abbreviation means.

Theoretical contributions

7. P21, “Firstly, while most of the existing research…on tourist satisfaction and willingness to revisit”. This sentence lacks citations and is not convincing enough.

6. PLOS authors have the option to publish the peer review history of their article (what does this mean?). If published, this will include your full peer review and any attached files.

Reviewer #1: No

Reviewer #2: No

---

## [Author Response · Author response to Decision Letter 0]

17 Jul 2023

Dear Editors and Reviewers:

Thank you for your letter and for the reviewers' comments concerning our manuscript entitled “An investigation of how perceived smart tourism technologies affect tourists' well-being in marine tourism” (ID: PONE-D-23-12482). Those comments are all valuable and very helpful for revising and improving our paper, as well as the important guiding significance to our researches. We have studied comments carefully and have made correction which we hope meet with approval. Revised portion are marked in yellow in the paper. The main corrections in the paper and the responds to the comments from editors and reviewers are as flowing:

Responds to additional editor comments:

I find your studies involved children and hope the required ethical considerations are taken care. Theoretical and managerial contribution need to be improved by citing the previous research done on this topic.

Response:

Given that it would have been inappropriate for the study to investigate minors, we have excluded samples aged under 18 years. We have added citations to the literature in the theoretical and managerial contributions section.

Responds to the reviewer’s comments:

Reviewer #1: 

1. Response to comment: The author mentions the application of smart tourism technology in different tourism domains. Could the author provide some specific examples through the literature? Why is there a special focus on marine tourism in the context of the tourism field? What is the main gap or rationale for this focus?

Response: We have added in the introduction section examples of the application of smart tourism technologies in different tourism sectors and why we should pay special attention to marine tourism.

2. Response to comment: In the section discussing hypothesis testing, the author briefly mentions literature on the impact of smart tourism technology on tourist well-being. Could the author provide further elaboration on why they are exploring the relationship between these two factors? Did they consider other potential confounding variables during the process? For example, does the attractiveness of the destination or personal characteristics of tourists influence well-being?

Response: We have added further to the literature on the impact of smart tourism technologies on tourists' well-being and the implications of exploring the relationship between these two factors in the hypotheses section. A few of the studies that have been conducted by scholars have considered other potential confounding variables, such as the perceived value of the destination also affecting well-being. 

3. Response to comment: Regarding the research on tourist well-being mentioned in the paper, is there any further explanation or differentiation of different types of well-being and their relationship with the tourist experience?

Response: We did not provide further explanations or distinctions between the different types of well-being and their relationship to the tourism experience. “The findings of this study indicate that tourists can experience an improvement in both their hedonic and eudaimonic well-being through the creation of memorable travel experiences. Memorable tourism experiences can lead to short-term hedonic well-being among visitors, as well as long-term increases in self-worth and eudaimonic well-being.”

4. Response to comment: As mentioned in this study, previous tourism literature has already discussed the relationship between smart technology and well-being. This study specifically investigates marine tourism. However, in the hypothesis and results discussion, it is not clear why marine tourism was chosen and there is a lack of literature discussion and traceability regarding marine tourism. This might weaken the contribution of this article. It is recommended to strengthen the literature review and discussion related to marine tourism.

Response: It is really true as Reviewer suggested that in the hypothesis and results discussion, it is not clear why marine tourism was chosen and there is a lack of literature discussion and traceability regarding marine tourism. Considering the Reviewer’s suggestion, we have strengthened the literature review and discussion related to marine tourism. Special thanks to you for your good comments.

Reviewer #2:

1. Response to comment: P3, “Although hedonic well-being with … personal potential, virtues, character strengths, self-growth, self-fulfillment”. The authors only describe which aspects it pays more attention to in the few tourism literatures that study eudaimonic well-being, but do not discuss the gaps between this study and existing research.

 Response: We are very sorry for our negligence of discussing the gaps between this study and existing research. We have added the gap between this study and existing research in the introduction section.

2. Response to comment: P9-P10, “The integration of smart tourism technologies…. In light of this, the following hypotheses were developed”. The discussion on the relationship between perceived smart tourism technology and tourist well-being in this paragraph is relatively weak. The examples given by the authors do not well support the hypothesis.

Response: We have strengthened the discussion of the relationship between perceived smart tourism technologies and tourist well-being in our Literature review section.

3. Response to comment: P13-P14, “This paper draws on previously validated scales…are based on research conducted by Diener”. Reviewers cannot see the scale for this study. It is recommended to show the complete scale.

Response: Initially, the complete scale was placed in the supporting information considering its relatively large size, and we have shown it in the manuscript for ease of reading.

4. Response to comment: P14, “This study is aimed at people…Table 1 lists more details of the sample for this study”. Why did the authors use a combination of online and offline methods to collect data? After collecting these two parts of the data, was there a t-test performed? How are offline questionnaires collected?

 Response: To be honest, the use of a combination of online and offline methods to collect the questionnaires was to increase the sample size as far as it was possible. A t-test has been done on the data. The offline questionnaire was collected by travelling to marine tourism destinations such as Shanghai Changfeng Ocean World, Shanghai Ocean Aquarium and Shanghai Haichang Ocean Park and presenting the QR code of the questionnaire to visitors who were willing to cooperate with the study.

5. Response to comment: P14, Table1: It is inappropriate to collect questionnaires from participants younger than 18 years of age.

 Response: As it would be inappropriate to collect questionnaires from participants under the age of 18, we excluded that part of the sample (44).

6. Response to comment: Table 2, Table 4 and Table 5: The authors did not explain what each abbreviation means.

 Response: We are very sorry for our negligence of explaining what each abbreviation means and we have added the meaning of each abbreviation below these three tables.

7. Response to comment: P21, “Firstly, while most of the existing research…on tourist satisfaction and willingness to revisit”. This sentence lacks citations and is not convincing enough.

 Response: This sentence was initially summed up by myself after reviewing a large amount of literature related to perceived smart tourism technology, and was deleted from the revised version considering that it lacked citations and was not persuasive enough. Special thanks for your valuable comments!

We tried our best to improve the manuscript and made some changes in the manuscript. These changes will not influence the content and framework of the paper. And here we did not list the changes but marked in yellow in revised paper.

We appreciate for Editors/Reviewers’ warm work earnestly, and hope that the correction will meet with approval.

Once again, thank you very much for your comments and suggestions.

---

## [Decision Letter · Decision Letter 1]

11 Aug 2023

An investigation of how perceived smart tourism technologies affect tourists' well-being in marine tourism

PONE-D-23-12482R1

Dear Dr. Wu,

We’re pleased to inform you that your manuscript has been judged scientifically suitable for publication and will be formally accepted for publication once it meets all outstanding technical requirements.

Kind regards,

Sudarsan Jayasingh, Ph.D

Academic Editor

PLOS ONE

---

## [Editor Report · Acceptance letter]

18 Aug 2023

PONE-D-23-12482R1 

An investigation of how perceived smart tourism technologies affect tourists' well-being in marine tourism 

Dear Dr. Wu:

I'm pleased to inform you that your manuscript has been deemed suitable for publication in PLOS ONE. Congratulations! Your manuscript is now with our production department. 

Kind regards, 

on behalf of

Dr. Sudarsan Jayasingh 

Academic Editor

PLOS ONE